# Aortic Root Downward Movement as a Novel Method for Identification of an Adequately Performed Valsalva Maneuver to Detect Patent Foramen Ovale during Transesophageal Echocardiograph

**DOI:** 10.3390/diagnostics12040980

**Published:** 2022-04-13

**Authors:** Lixin Chen, Yuanyuan Sheng, Yuxiang Huang, Jian Li, Xiaohua Liu, Qian Liu, Bobo Shi, Xiaofang Zhong, Jinfeng Xu, Yingying Liu

**Affiliations:** Shenzhen Medical Ultrasound Engineering Center, Department of Ultrasound, Shenzhen People’s Hospital (The Second Clinical Medical College, Jinan University; The First Affiliated Hospital, Southern University of Science and Technology), Shenzhen 518020, China; neostar84@aliyun.com (L.C.); shengyyah@163.com (Y.S.); ln4162643454@163.com (Y.H.); dr_lijian@126.com (J.L.); lengfeng0418@163.com (X.L.); zctsdzct@163.com (Q.L.); sumi150117@163.com (B.S.); zxflds1988@163.com (X.Z.)

**Keywords:** patent foramen ovale, transesophageal echocardiography, aortic root downward movement, Valsalva maneuver

## Abstract

The Valsalva maneuver (VM) is the most sensitive auxiliary method for the detection of patent foramen ovale (PFO), but it is difficult to assess whether the maneuver is adequately performed during transesophageal echocardiography (TEE). In this study, we tried to use aortic root downward movement as a novel method for judging whether VM was adequate or not, and to evaluate whether this novel method can increase the sensitivity of detecting PFO. A total of 224 patients with clinically suspected PFO were enrolled in this study. These patients were injected with activated normal saline to detect the right-to-left shunt (RLS), in the following three conditions: contrast-enhanced transthoracic echocardiography under adequate VM (AVM cTTE), contrast-enhanced TEE under non-adequate VM (non-AVM cTEE), and cTEE under adequate VM (AVM cTEE). A novel method in which the aorta root moves downward (movement range ≥16 mm) in the cTEE judged whether AVM was performed. The PFO detection rate and sensitivity of AVM cTEE were better than that of non-AVM cTEE (detection rate: 108 PFOs [48.2%] vs. 86 PFOs [38.4%], *p* = 0.036; sensitivity: 100% vs. 79.6%). Among AVM cTTE, non-AVM cTEE, and AVM cTEE, the RLS grade evaluation results were inconsistent, with significant differences (*p* < 0.05). Non-AVM cTEE had RLS underestimation or false negatives. Compared with non-AVM cTEE, AVM cTEE and AVM cTTE had better consistency in evaluating PFO RLS (kappa value = 0.675). Aortic root downward movement could be used as a novel method for judging the effectiveness of VM, which is critical for the detection of PFO in cTEE. Concerning effectiveness and convenience, this method should be promoted during the clinical detection of PFO.

## 1. Introduction

Patent foramen ovale (PFO) is an important cause of cryptogenic stroke, especially in young patients whose morbidity is as high as 46% [1]. Even in ordinary adults, the morbidity of PFO is approximately 25% [2]. Domestic and foreign research in the field of PFO and cryptogenic stroke is progressing rapidly. Internationally, four randomized controlled trials [3,4,5,6] have shown that in reducing the risk of stroke recurrence, PFO closure is better than drug therapy alone. With the expansion of PFO treatment methods, efficiently and accurately increasing the detection rate of PFO has a significant meaning in clinical practice.

Although right heart catheterization and the confirmation of the guidewire passing through the septum are the most accurate ways to confirm the presence of PFO, considering its invasiveness, it is not feasible to perform right heart catheterization to diagnose PFO [7]. At present, many studies believe that contrast-enhanced transesophageal echocardiography (cTEE) under adequate Valsalva maneuvers (AVMs) is the gold standard for diagnosing PFO [2,8,9]. Compared with transthoracic echocardiography (TTE), transesophageal echocardiography (TEE) can more intuitively display the anatomical structure of the foramen ovale in the atrial septum, determine the source of the shunt from the foramen ovale or pulmonary vein, and quantify the size of the shunt tube [10]. Unlike TTE, in which AVM was clearly determined by the subject in supine or semisupine position keeping the mercury column of the sphygmomanometer at a position of 40 mm for 10 s [11], TEE cannot replicate this method for VM judgment because of oral intubation. According to existing research, there is still no unified and effective method for judgment of AVM [11,12,13]. Even worse, compared with TTE, there were more patients who could not effectively cooperate with VM during TEE, resulting in false-negative results of PFO detection. Therefore, we tried to find a new method for judging AVM in cTEE examination.

As we know, when the VM is performed, the chest pressure rises, the diaphragm decreases, and the mediastinum structures such as the heart and the root of the aorta will move downward. We have also found this phenomenon in clinical TEE operations, that is, when patients perform VM, the downward movement of the heart and aortic root can be observed. Compared with the heart, the movement of the aortic root is easier to locate and measure. Therefore, in this study, we observed whether it was effective to use aortic root downward movement in the cTEE for determining AVM and further for detecting PFO.

## 2. Materials and Methods

### 2.1. Patient Population

A total of 233 patients with clinically highly suspected PFO were examined for cTTE and cTEE from May 2018 to December 2020. The exclusion criteria were atrial fibrillation, significant heart disease including moderate and severe valvular disease, moderate and severe left ventricular (LV) systolic dysfunction, congenital heart disease except for PFO, and other cardiac sources of embolism such as thrombi, tumors, or vegetation. Before the echocardiographic examination, each subject needed to be trained in the Valsalva movement. The sign of AVM is: use a pressure gauge to measure the pressure and increase the chest pressure by ≥40 mmHg (1 mmHg = 0.133 kPa) [11,14]. Eight cases in which the Valsalva maneuver was unable to be performed in the echo laboratory before echocardiography and 1 case diagnosed as an atrial septal defect by TEE were initially excluded. Finally, 224 cases were enrolled in this study. All cases showed a sinus heart rate during the examination, and all patients gave written informed consent for non-sedation TEE and TTE using agitated saline contrast injection.

In accordance with the established guidelines of the American Society of Echocardiography [2,15], each subject was subjected to routine TTE and comprehensive TEE evaluations, including two-dimensional echocardiography and color Doppler. The TTE and TEE measurement instrument was a Phillip EPIQ7C system (Philips Ultrasound, Bothell, WA, USA) equipped with an S5-1 probe, frequency 1–5 MHz, and an X7-2T probe, frequency 2–7 MHz. For local anesthesia of the oropharynx, 2% lidocaine gel was used without the need for intravenous sedatives. The saline contrast was produced by 1 mL air, 1 mL blood, and 8 mL saline [2]. The contents were stirred between two 10 mL syringes connected to the tee and quickly injected from the anterior elbow vein. The following three conditions were compared: cTTE under adequate Valsalva maneuver (AVM cTTE), cTEE under non-adequate Valsalva maneuver (non-AVM cTEE), and cTEE under adequate Valsalva maneuver (AVM cTEE). A novel method in which the aorta root moves downward (movement range ≥ 16 mm, which was determined according to our pre-experiment, and will be described in detail later) in the cTEE examination judged whether AVM was performed. Each patient underwent two or more VMs during the cTEE examination, of which at least one was judged to be adequate VM: cTEE with aortic root downward movement group (AVM cTEE), at least one was judged to be non-AVM: cTEE without aortic root downward movement group (non-AVM cTEE). In the cTTE examination, a pressure gauge was used to measure the pressure, and the pleural pressure increase of ≥40 mmHg (1 mmHg = 0.133 kPa) was judged to be adequate in the VM:cTTE group (AVM cTTE). When the results of the TEE examination were not satisfactory, repeated right heart angiography was performed, and the subject was guided to fully complete the VM until aortic root downward movement during the TEE examination, especially in the case of negative RLS. cTTE selected the apical four-chamber view for observation, and cTEE selects images of the interatrial septum that were obtained from the best imaging plane for septal membrane visualization, typically 60° to 90°. The number of microbubbles in the left heart was observed within 3 cardiac cycles after the right heart was developed after effective VM at rest to determine the amount of RLS.

### 2.2. Definitions

During the cTEE examination, when the subject performed the VM, TEE observed the interatrial septum (IAS) on the fossa ovale cut surface to see that the aorta root downward moved from the resting position ≥16 mm, and at the end of the VM moved up to the original resting position, which was defined as the downward movement of the aorta root during the cTEE examination and was judged to be an AVM (Figure 1A,B). When the subject performed the VM, TEE observation of the IAS on the fossa ovale cut surface showed that the aorta root downward moved from the resting position <16 mm or did not move, which was defined as the absence of downward movement of the aorta root during the cTEE examination and judged to be a non-AVM. During the TEE examination, professional and experienced nurses checked the contraction of the subject’s abdominal muscles to determine the subject’s VM.

A pre-experiment was conducted to define the cutoff value of aortic root downward movement. Fifty persons of different ages and genders who underwent TTE in our hospital were randomly selected. During the examination, the pressure gauge method to measure the pressure and increase the chest pressure by ≥40 mmHg, and the position changes of the aorta root were observed. In order to correspond to the TEE section, after the previous multi-section test, a non-standard view, which was obtained from a slightly skewed the short-axis view of the great artery showing the atrial septum, was finally selected as the transthoracic observation section of aortic root downward movement (Figure 1C,D). The median of these values was selected as the cut-off value of aortic root downward movement in this study (Figure 2).

When using agitated saline contrast injection to see at least one clear microbubble in the left atrial (LA) or LV in the recorded movie, the diagnosis was positive for RLS. The number of microbubbles in LA (TEE) and LV (TTE) is counted in a single still frame showing the maximum number of microbubbles, and the amount of RLS is determined by the number of microbubbles appearing in the left ventricular cavity on the still single frame image. Quantification of LA opacification was regarded as grade 0 (zero microbubbles), grade 1 (mild; 1 to 10 microbubbles), grade 2 (moderate; 11 to 30 microbubbles) or grade 3 (severe; >30 microbubbles) [16,17].

### 2.3. Demographic Data Collection

Clinical data included age, sex, body surface area, body mass index, migraine headache, atrial septal aneurysm, documented diagnosis of hypertension, transient ischemic attack, and stroke (Table 1).

### 2.4. Observer Variability

RLS in PFO and the consistency between observers and within observers were evaluated. cTTE, non-AVM cTEE, and AVM cTEE were also evaluated to determine the degree of RLS classification and the consistency between observers, within observers, and between the three conditions.

### 2.5. Statistical Analysis

Categorical variables are presented as a number and percentage (%). Continuous variables are expressed as the mean ±SD. We applied the chi-square test to compare the rates of detection. AVM cTEE is used as the gold standard for the diagnosis of PFO, comparing the sensitivity of AVM cTTE, non-AVM cTEE, and AVM cTEE. The Friedman test was used to analyze the data of the quantification of the shunt, and then we compared the results of each examination with those by the Wilcoxon matched pairs test (*p* values of <0.0167 were considered statistically significant (with the correction of Bonferroni)). The consistency between observers, within observers, and between two different inspection methods was tested by Kappa. *p* < 0.05 was considered to indicate statistical significance. Statistical analyses were performed using IBM SPSS V.20.0 software.

## 3. Results

### 3.1. Comparisons of PFO Detection Rate and Sensitivity

All 224 subjects with suspected PFO underwent cTTE and cTEE examinations, of which 108 patients (48.2%) were diagnosed with PFOs by AVM cTEE, 86 patients (38.4%) by non-AVM cTEE, and 100 patients (44.6%) by AVM cTTE. The PFO detection rate of AVM cTEE washigher than non-AVM cTEE (*p* = 0.036). The sensitivity of detecting PFO for AVM cTTE, non-AVM cTEE, and AVM cTEE was 92.6%, 79.6%, and 100%, respectively.

### 3.2. Semiquantitative Shunt Grading

The number of cases at each level detected by the three conditions ispresented in Table 2. When AVM cTTE and AVM cTEE PFO werepositive, RLS wasmostly grade 3, while non-AVM cTEE wasmostly grade 1 or even false negative. Figure 3 shows that one patient had RLS Grade 3 in AVM cTTE and AVM cTEE but RLS Grade 1 in non-AVM cTEE. In shunt quantification, there was a statistically significant difference between the three conditions (*p* < 0.05), and a statistically significant difference was evident from the results between AVM cTEE and non-AVM cTEE (*p* < 0.01), between AVM cTTE and non-AVM cTEE (*p* < 0.01), and between AVM cTTE and AVM cTEE (*p* > 0.01).

Figure 4 shows three different conditions: the size of each RLS grade evaluated below and the corresponding relationship between the RLS grade in different conditions. For example, there were 22 cases of non-AVM cTEE RLS grade 0, but in AVM cTEE, there were 11 cases of RLS grade 1, 4 cases of RLS grade 2, and 7 cases of RLS grade 3.

### 3.3. Consistency of Judging RLS Grade in Three Different Conditions

AVM cTEE and AVM cTTE have better consistency of RLS (k = 0.675; *p* = 0.038). The consistency of the other two groups was poor (k = 0.369, *p* = 0.030, cTTE and non-AVM cTEE; k = 0.386, *p* = 0.033, non-AVM cTEE and AVM cTEE).

### 3.4. Observer Variability

AVM cTTE showed good intraobserver and interobserver agreement when judging the RLS grade (both k = 0.854, *p* < 0.0001, agreement, 93.3%). AVM cTEE had good intraoberver and interoberver agreement when judging the RLS grade (k = 0.896, *p* < 0.0001, agreement, 93.3%; k = 0.947, *p* < 0.0001, agreement, 96.7%); non-AVM cTEE had good intraoberver and interoberver agreement when judging the RLS grade (k = 0.884, *p* < 0.0001, agreement, 93.3%; k = 0.940, *p* < 0.0001, agreement, 96.7%); 100% consistency was used in determining whether there was RLS between the intraobserver and interobserver groups in 3 conditions (AVM cTTE, non-AVM cTEE and AVM cTEE).

## 4. Discussion

VM is an easy-to-perform maneuver that involves forced expiration of the closed nose and mouth after a deep breath. During deep inhalation, the pressure in the thoracic cavity increases, and part of the venous return is blocked, which reduces the blood flow of the right heart back to the heart. The right heart preload decreased, and the right heart pressure decreased. During deep exhalation, thoracic pressure decreases, and a sudden increase in venous return causes a sudden increase in blood flow in the right heart and an increase in right heart preload and right heart pressure. The pressure in the right heart increases, resulting in a short-term increase in the pressure gradient between the right and left atrium. If PFO is present, it can show a right-to-left shunt signal. At present, in most studies, VM is still widely used in the diagnosis of PFO, and the importance of AVM for the detection of PFO is emphasized [18]. In previous studies [11,12,13], the method to judge whether VM is adequate was as follows: (1) the patient blows into the mouthpiece in the supine or semirecumbent position and maintains the column of mercury in the sphygmomanometer at 40 mm for a period of ten seconds. (2) Adequate Valsalva is evident by a decrease in mitral inflow peak E velocity of 20 cm/s in the strain phase and (3) leftward shift of the interatrial septum in the release phase as an indication of right atrial (RA) pressure exceeding that of LA. First, the pressure gauge insufflation method can only be used for TTE because the TEE probe is not conducive to use for insufflation pressure measurement, so it cannot be used for TEE. E wave speed reduction and IAS shift can be used in TTE and TEE inspections to determine the effectiveness of VM. The decrease in E wave velocity can reflect the decrease in right heart preload caused by the decrease in venous return, thereby judging that an AVM has been performed. The disadvantage is that it cannot be judged by TEE at the same time and on the same plane. The effectiveness of the VM cannot be simply equal in two different aspects under different time phases. The leftward shift of the interatrial septum in the release phase is a convenient and simple judgment method that can be visually observed in real-time. Its disadvantage is that the leftward shift of IAS mainly occurs at the moment when the VM ends, and the validity cannot be judged in time during the inspection process. Existing research still lacks a unified judgment method for AVM in TEE.

Ways of improving the sensitivity and specificity of TEE in the diagnosis of PFO havealways been a field worthy of study. TEE is considered the reference standard for the detection of a PFO in the 2015 ASE guidelines and recent studies. Under normal conditions, whether using TTE or TEE, the accuracy of the test will be improved by the use of multiple injections of agitated saline with provocative maneuvers to transiently increase the RA pressure, including the adequate Valsalva maneuver (VM) and cough [19]. If the patient is under anesthesia state, the VM can be mimicked by held inspiration and then release, conventional abdominal compression, or inferior vena cava (IVC) compression maneuver [2,20,21]. However, there is no consensus in the current research on the evaluation of VM effectiveness in TEE.

Combining our experience and thinking in clinical practice, in this study, we tried to use aortic root downward movement as a new method for judging whether VM is adequate or not in TEE examination. AVM will cause the diaphragm to move downward due to the increased pressure in the thoracic cavity during deep inhalation. When the diaphragm moves downward, the position of the heart and aortic root will also move downward, as seen in the TEE section. At that time, part of the reflux of the inferior vena cava into the right atrium isblocked, and the right heart preload isreduced; after deep exhalation, the sudden drop in thoracic pressure causes the diaphragm to move up, and the aorta root also returns to its original position. At this time, the sudden increase in venous return causes the right-to-left shunt. The increase in precardiac load results in a transient increase in the pressure gradient between the right and left atrium. If PFO is present, it can show the signal of right-to-left shunt microbubbles, thereby verifying the diagnosis of PFO. The downward movement of the diaphragm during AVM cannot be observed on TEE, while the accompanying obvious downward movement of the aorta root can be observed on TEE, and it can be achieved in the PFO diagnostic view. Therefore, it is feasible in principle to judge whether the patient has performed AVM by aortic root downward movement. In this study, we judged the Valsalva maneuver as AVM and non-AVM in cTEE according to whether there was aortic root downward movement.

In the current study, we compared the PFO detection rate and sensitivity of AVM cTTE, non-AVM cTEE, and AVM cTEE. The effects of different conditions on the grade of PFO RLS were observed. Many previous studies have compared the use of cTEE as the gold standard or the use of cTTE- or cTEE-confirmed PFO diagnosis to compare the accuracy of TTE and TEE in diagnosing intracardiac shunts. These studies show different conclusions. Thanigaraj and Yue L et al. showed that the sensitivity of TTE to TTE or TEE diagnosed with PFO is higher than that of TEE (the sensitivity to TTE is 100%, the sensitivity to TEE is 86%, the sensitivity to TTE is 86%, and the sensitivity to TEE is 56%) [22,23]. Another study showed that compared with TEE, TTE has good consistency in diagnosing RLS (99% sensitivity and 85% specificity) [24]. Other studies have shown that compared with TEE, it shows low sensitivity and high specificity for TTE (68% and 93% specificity, respectively; 46% and 99% specificity) [25,26]. These different results may be caused by different methods and different standards. In previous studies, the low sensitivity of TEE seems to be related to ineffective VM and sedation during TEE. In this study, it can be seen that the detection rate and sensitivity of PFO of AVM cTEE are higher than AVM cTTE (48.2% detection rate and 100% sensitivity vs. 44.6% detection rate and 92.6% sensitivity) and non-AVM cTEE was lower than AVM cTTE (38.4% detection rate and 79.6% sensitivity vs. 44.6% detection rate and 92.6% sensitivity).

In patients with PFO-associated stroke, the presence of a large shunt or atrial septal aneurysm has been suggested to convey high risk of stroke recurrence [27]. The results indicate that in patients with cryptogenic stroke, PFO closure can reduce the risk of recurrent strokes by almost 61% compared to standard medical therapy. It is important to note that the risk reduction was dependent on the shunt size, especially in those with moderate-to-large shunts, and patients with small shunts did not appear to benefit from PFO closure. Hence, shunt size should be considered when evaluating patients for potential PFO closure for the prevention of recurrent stroke [28]. In this study, when the VM was adequate, the higher the PFO RLS grade was judged, it was mostly RLS grade 3, while the invalid or insufficient VM judged the RLS classification to be grade 1 or false negatives. The two groups had poor consistency in diagnosing RLS. At the same time, this study also pointed out that compared with non-AVM cTEE, AVM cTEE and AVM cTTE have better consistency in diagnosing RLS. In TTE, insufflation with a pressure gauge under sedation can ensure the effectiveness of VM very well and accurately. TEE shows that the aorta root moves downward, which also proves the AVM to a certain extent. There was better than no aortic root downward shift, which was consistent with the results of this study. AVM cannot only improve the detection rate of PFO but also make a semiquantitative assessment of RLS more accurate.

The current study presents several limitations. First, this study was conducted in a single center and needs to be further verified by a large-scale multicenter study. Second, this study used an echocardiographic machine with the same image setting and vendor, which may prevent us from generalizing the results to other centers using different machines. third, there was selection bias because the subjects of this study were people who were clinically highly suspected of PFO and underwent adequate Valsalva maneuvers before echocardiography. Finally, because not all PFO positive subjects underwent PFO closure in this study, and the right heart catheterization was an invasive operation, cTEE with AVM was selected as the gold standard for the diagnosis of PFO.

## 5. Conclusions

Adequate to the Valsalva maneuver (with aortic root downward movement), cTEE has a higher disease detection rate and sensitivity to PFO. Compared with AVM cTEE and AVM cTTE, non-AVM cTEE assesses RLS grade as a small shunt or false negatives, and AVM cTEE and AVM cTTE have better consistency in assessing the degree of PFO RLS. Aortic root downward movement (range of movement ≥16 mm) can be used as a novel method for cTEE to determine an AVM, which is simple and convenient, can be visually judged in real-time on the same view of ultrasound, and can be guided in real-time during the TEE inspection process, thereby further improving the detection rate of PFO and the accuracy of RLS grade, which is helpful to guide the diagnosis and treatment of clinical PFOs.

## Figures and Tables

**Figure 1 diagnostics-12-00980-f001:**
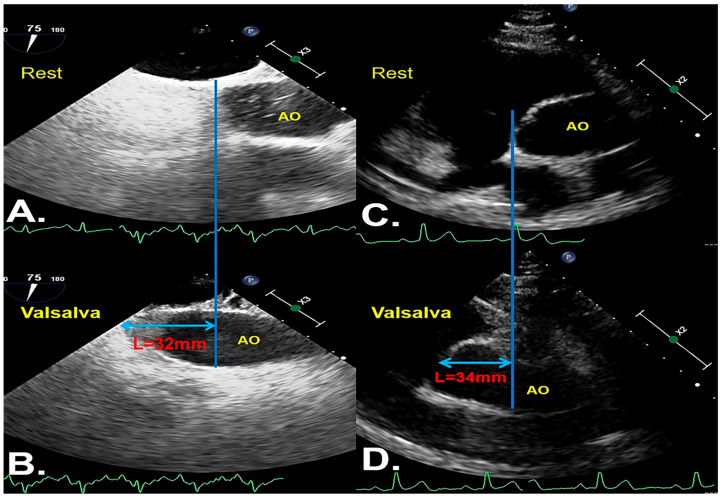
(**A**,**B**) Position images of the same VM at different time phases in cTEE. (**C**,**D**) Position images of the same VM at different time phases in TTE. (**A**,**C**) show the position of the aorta displayed in the resting state. (**B**,**D**) Position images when the aorta moved down, which was most obvious during VM. The blue line indicates that the aortic annulus is horizontally downward as a vertical line under the resting state, and this vertical line is the baseline. The light blue double arrow indicates the distance L that the aorta moves down beyond the baseline in the examination. AVM, adequate Valsalva maneuver; TTE, transthoracic echocardiography; cTEE, contrast-enhanced transesophageal echocardiography; AO; aorta.

**Figure 2 diagnostics-12-00980-f002:**
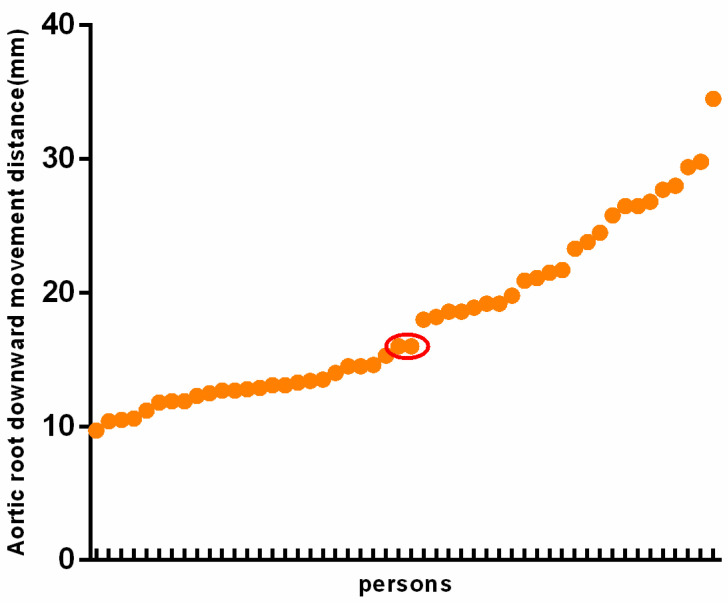
Data of aortic root downward movement in transthoracic echocardiographic in pre-experiment. The red elliptical ring is median.

**Figure 3 diagnostics-12-00980-f003:**
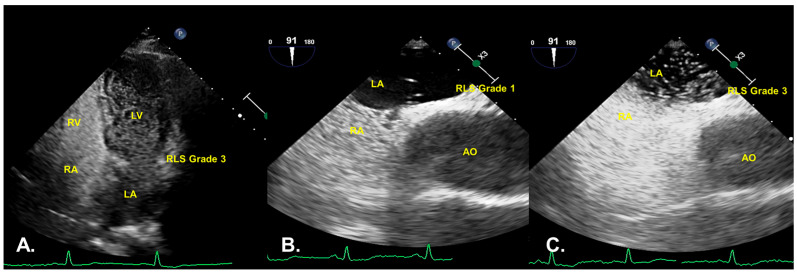
The grade of the PFO RLS shunt in the case of AVM cTTE (**A**), non-AVM cTEE (**B**), and AVM cTEE (**C**) in the same patient. RLS Grade 3 in AVM cTTE, RLS Grade 1 in non-AVM cTEE, RLS Grade 3 in AVM cTEE.LV, left ventricular; LA, left atrial; RA, right atrial; RV, right ventricular; AO; aorta; RLS, right-to-left shunt; PFO, patent foramen ovale; AVM, adequate Valsalva maneuver; cTTE, contrast-enhanced transthoracic echocardiography; cTEE, contrast-enhanced transesophageal echocardiography.

**Figure 4 diagnostics-12-00980-f004:**
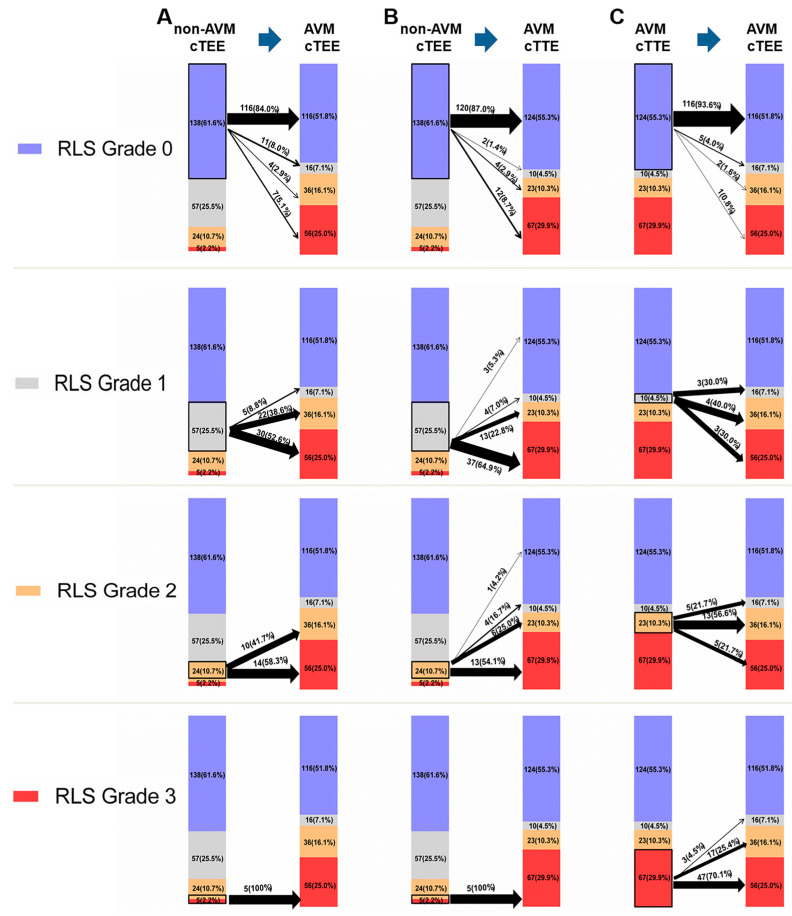
Reclassification between each RLS assessed by different conditions to determine RLS Grade 0, Grade 1, Grade 2, and Grade 3. The widths of arrows identify reclassification rates. (**A**) Initial classification by non-AVM cTEE reclassified by AVM cTEE. (**B**) Initial classification by non-AVM cTEE reclassified by AVM cTTE. (**C**) Initial classification by AVM cTTE reclassified by AVM cTEE. RLS, right-to-left shunt; AVM, adequate Valsalva maneuver; cTTE, contrast-enhanced transthoracic echocardiography; cTEE, contrast-enhanced transesophageal echocardiography.

**Table 1 diagnostics-12-00980-t001:** Patient characteristics of the study population.

Patient Characteristic	Value
Age (y)	44 ± 14
Men/women	92/132
Body surface area (m^2^)	1.68 ± 0.14
Body mass index (kg/m^2^)	23.3 ± 2.5
Migraine headache, n (%)	122 (54)
Medical history, n (%)	
Hypertension	27 (12)
Transient ischemic attack	5 (2.2)
Stroke	52 (23)
Atrial septal aneurysm	4 (1.8)

Data are expressed as mean ± SD or as number (percentage).

**Table 2 diagnostics-12-00980-t002:** The semiquantitative grading of right-to-left shunt under three conditions.

State	Right-to-Left Shunt			
	Grade 0	Grade 1	Grade 2	Grade 3
AVM cTTE	124 (55.3%)	10 (4.5%)	23 (10.3%)	67 (29.9%)
non-AVM cTEE	138 (61.6%)	57 (25.5%)	24 (10.7%)	5 (2.2%)
AVM cTEE	116 (51.8%)	16 (7.1%)	36 (16.1%)	56 (25.0%)

cTTE, contrast-enhanced transthoracic echocardiography; cTEE, contrast-enhanced transesophageal echocardiography; AVM: adequate Valsalva maneuver.

## Data Availability

The data presented in this study are available on request from the corresponding author.

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
