# Peer review of "Aortic Root Downward Movement as a Novel Method for Identification of an Adequately Performed Valsalva Maneuver to Detect Patent Foramen Ovale during Transesophageal Echocardiography"

_diagnostics, 2022, doi:10.3390/diagnostics12040980_

Round 1
Reviewer 1 Report
Dear authors,
Your paper submitted for evaluation is a nice work but no relevant new issues are proposed. There are reviews and state-of-the-art papers in this field.
Why do You consider 2D TOE instead of more precisely 2D TOE?
Please explain why the saline used contains blood? It is atypical.
Please expand the chapter discussions.
The refferences must be impoved with refference papers in this field.
The manuscript must be again revised for spelling, grammar maybe in a proffesional service.
Author Response
Response to Reviewer 1 Comments
Point 1: Your paper submitted for evaluation is a nice work but no relevant new issues are proposed. There are reviews and state-of-the-art papers in this field.
Response 1: Thank you for your comments. According to your suggestion, I consulted the latest reviews and papers in this field, added the discussion part of the manuscript , and updated the references in the manuscript,which is highlighted in red on Page 8 (L266-275)and refferences of the revised manuscript.
Point 2: Why do You consider 2D TOE instead of more precisely 2D TOE?
Response 2: TEE is considered the reference standard for detection of a PFO in the 2015 ASE guidelines and recent studies. Whether using TTE or TEE, the accuracy of the test will be improved by the use of a standardized protocol that includes multiple injections of agitated saline with provocative maneuvers to transiently increase the RA pressure. Therefore, cTEE under adequate Valsal-vamaneuvers (AVMs) is the gold standard for diagnosing PFO in this manuscript, which is highlighted in red on Page 2(L51-53)of the revised manuscript.
- Silvestry, F.E.; Cohen, M.S.; Armsby, L.B.; Burkule, N.J.; Fleishman, C.E.; Hijazi, Z.M.; Lang, R.M.; Rome, J.J.; Wang, Y.; American Society of, E.; et al. Guidelines for the Echocardiographic Assessment of Atrial Septal Defect and Patent Foramen Ovale: From the American Society of Echocardiography and Society for Cardiac Angiography and Interventions. J Am Soc Echocardiogr 2015, 28, 910-958, doi:10.1016/j.echo.2015.05.015.
- Yang, J.; Zhang, H.; Wang, Y.; Zhang, S.; Lan, T.; Zhang, M.; Li, Y.; Huang, W.; Zhang, H.; Wang, A.; et al. The Efficacy of Contrast Transthoracic Echocardiography and Contrast Transcranial Doppler for the Detection of Patent Foramen Ovale Related to Cryptogenic Stroke. Biomed Res Int 2020, 2020, 1513409, doi:10.1155/2020/1513409.
- Zhao, E.; Du, Y.; Xie, H.; Zhang, Y. Modified Method of Contrast Transthoracic Echocardiography for the Diagnosis of Patent Foramen Ovale. Biomed Res Int 2019, 2019, 9828539, doi:10.1155/2019/9828539.
Point 3: Please explain why the saline used contains blood? It is atypical.
Response 3: This is a good question.According to the 2015 ASE guidelines, the agitated saline which contains blood (8 mL of saline plus 1 mL of blood from the patient plus 1 mL air) was recommended. Meanwhile,We found that the addition of blood to the contrast solution result in increased intensity of the microbubbles detected by echocardiography in clinical practice. Besides, we use the examiner's own blood, so there is no ethical problem. We added corresponding references to the manuscript. Page 2 (L94-95)
- Silvestry, F.E.; Cohen, M.S.; Armsby, L.B.; Burkule, N.J.; Fleishman, C.E.; Hijazi, Z.M.; Lang, R.M.; Rome, J.J.; Wang, Y.; American Society of, E.; et al. Guidelines for the Echocardiographic Assessment of Atrial Septal Defect and Patent Foramen Ovale: From the American Society of Echocardiography and Society for Cardiac Angiography and Interventions. J Am Soc Echocardiogr 2015, 28, 910-958, doi:10.1016/j.echo.2015.05.015.
Point 4: Please expand the chapter discussions.
Response 4: According to your suggestion, the methods of evaluating adquate VM in the current study are described in detail in this manuscript, which is highlighted in red on Page 8 (L247-264).
we added a discussion section to the manuscript, which is highlighted in red on Page 8 (L266-275)of the revised manuscript.
The additional discussion is as follows:
How to improve the sensitivity and specificity of TEE in the diagnosis of PFO has always been a field worthy of study. TEE is considered the reference standard for the detection of a PFO in the 2015 ASE guidelines and recent studies. Under normal condi-tions, whether using TTE or TEE, the accuracy of the test will be improved by the use of multiple injections of agitated saline with provocative maneuvers to transiently in-crease the RA pressure,including the adequateValsalva maneuver (VM) and cough[1]. If the patient is under anesthesia state, the VM can be mimicked by held inspiration and then release, conventional abdominal compression, or inferior vena cava( IVC) compression maneuver[2-4]. However, there is no consensus in the current research on the evaluation of VM effectiveness in TEE.
- Wang, S.B.; Wang, X.C.; Ma, Y.; Liu, K.D.; Xing, Y.Q. Right-to-left shunt detection using contrast-enhanced transcranial Doppler: A comparison of provocation maneuvers between coughing and a modified Valsalva maneuver. PLoS One 2017, 12, e0175049, doi:10.1371/journal.pone.0175049.
- Takaya, Y.; Watanabe, N.; Ikeda, M.; Akagi, T.; Nakayama, R.; Nakagawa, K.; Toh, N.; Ito, H. Importance of Abdominal Compression Valsalva Maneuver and Microbubble Grading in Contrast Transthoracic Echocardiography for Detecting Patent Foramen Ovale. Journal of the American Society of Echocardiography 2020, 33, 201-206, doi:10.1016/j.echo.2019.09.018.
- Yamashita, E.; Murata, T.; Goto, E.; Fujiwara, T.; Sasaki, T.; Minami, K.; Nakamura, K.; Kumagai, K.; Naito, S.; Kario, K.; et al. Inferior Vena Cava Compression as a Novel Maneuver to Detect Patent Foramen Ovale: A Transesophageal Echocardiographic Study. J Am Soc Echocardiogr 2017, 30, 292-299, doi:10.1016/j.echo.2016.11.011.
- Silvestry, F.E.; Cohen, M.S.; Armsby, L.B.; Burkule, N.J.; Fleishman, C.E.; Hijazi, Z.M.; Lang, R.M.; Rome, J.J.; Wang, Y.; American Society of, E.; et al. Guidelines for the Echocardiographic Assessment of Atrial Septal Defect and Patent Foramen Ovale: From the American Society of Echocardiography and Society for Cardiac Angiography and Interventions. J Am Soc Echocardiogr 2015, 28, 910-958, doi:10.1016/j.echo.2015.05.015.
Point 5: The refferences must be impoved with refference papers in this field.
Response 5: Thank you for your positive suggestion. How to improve the sensitivity and specificity of TEE in the diagnosis of PFO has always been a field worthy of study. We have updated the references in the manuscriptand and cited more papers published in this field in recent 5 years,which is highlighted in red on refferences of the revised manuscript.
Point 6: The manuscript must be again revised for spelling, grammar maybe in a proffesional service.
Response 6: Thank you very much for your positive comments and kind suggestion. The authors have made improvements in this revised version, according to your kind suggestion. We have found professionals to polish the article before submission, and try again to make changes before committing this time. Our certificate is attached below. If reviewers think that English polishing is still needed, we will make further improvement.

Reviewer 2 Report
the work is very interesting, I suggest to edit the images with better quality
Author Response
回复审稿人 2 条评论
Point 1: 作品很有趣,我建议编辑质量更好的图像。
回应1:非常感谢您的积极评价和善意建议。我们已经编辑
图像的质量,包括提高图像的分辨率和美感。如果审稿人认为仍然需要更好的图像质量,我们将进一步改进。

Round 2
Reviewer 1 Report
Dear authors,
I received and read the revised version of your manuscript.
You made improvements in the paper.
Sincerely Yours,
Reviewer